# Security in Transformer Visual Trackers: A Case Study on the Adversarial Robustness of Two Models

**DOI:** 10.3390/s24144761

**Published:** 2024-07-22

**Authors:** Peng Ye, Yuanfang Chen, Sihang Ma, Feng Xue, Noel Crespi, Xiaohan Chen, Xing Fang

**Affiliations:** 1School of Cyberspace, Hangzhou Dianzi University, Hangzhou 310018, China; peng.ye@dbappsecurity.com.cn (P.Y.); sihangma1@gmail.com (S.M.); jhchxh0320@gmail.com (X.C.); fangxing724@gmail.com (X.F.); 2Key Laboratory of Discrete Industrial Internet of Things of Zhejiang, Hangzhou 310018, China; 3DBAPPSecurity Co., Ltd., Hangzhou 310051, China; 4ZJU-Hangzhou Global Scientific and Technological Innovation Center, Zhejiang University, Hangzhou 310058, China; henryxue@zju.edu.cn; 5Institut Polytechnique de Paris, Institut Mines-Telecom, 91120 Paris, France; noel.crespi@mines-telecom.fr

**Keywords:** autonomous driving, visual tracking, adversarial attacks, transformer model

## Abstract

Visual object tracking is an important technology in camera-based sensor networks, which has a wide range of practicability in auto-drive systems. A transformer is a deep learning model that adopts the mechanism of self-attention, and it differentially weights the significance of each part of the input data. It has been widely applied in the field of visual tracking. Unfortunately, the security of the transformer model is unclear. It causes such transformer-based applications to be exposed to security threats. In this work, the security of the transformer model was investigated with an important component of autonomous driving, i.e., visual tracking. Such deep-learning-based visual tracking is vulnerable to adversarial attacks, and thus, adversarial attacks were implemented as the security threats to conduct the investigation. First, adversarial examples were generated on top of video sequences to degrade the tracking performance, and the frame-by-frame temporal motion was taken into consideration when generating perturbations over the depicted tracking results. Then, the influence of perturbations on performance was sequentially investigated and analyzed. Finally, numerous experiments on OTB100, VOT2018, and GOT-10k data sets demonstrated that the executed adversarial examples were effective on the performance drops of the transformer-based visual tracking. White-box attacks showed the highest effectiveness, where the attack success rates exceeded 90% against transformer-based trackers.

## 1. Introduction

In recent years, autonomous vehicles have relied heavily on advanced sensor technologies, such as LIDAR, radar, GPS, and ultrasonic sensors, to navigate and understand their environments [1]. Cameras, as a significant part of this sensor suite, provide crucial visual data for tasks like target tracking, traffic sign recognition, and lane detection. This image data plays a pivotal role in understanding dynamic scenes and tracking moving objects for safe autonomous driving. However, reliance on image data also brings particular vulnerabilities. Visual target tracking, which primarily depends on this camera-based image data, has seen remarkable improvements with the advent of deep learning models, particularly transformer models. Models based on convolutional neural networks (CNNs) have made significant advancements in visual tracking. For example, Siamese networks (such as SiamFC [2], SiamRPN [3], and SiamMask [4]) primarily use CNNs for feature extraction. MDNet [5] employs a multi-domain training strategy, while ATOM [6] combines CNNs with bounding box prediction techniques. These models have improved the performance and robustness of object tracking to varying degrees. Additionally, models based on recurrent neural networks (RNNs) and long short-term memory networks (LSTMs) have also been applied to visual tracking. These models leverage RNNs or LSTMs to capture temporal dependencies in video sequences, sometimes in combination with CNNs for feature extraction, and excel at handling dynamic targets and temporal sequence data. However, transformer models have shown even better performance in visual tracking. Their remarkable ability to capture long-range dependencies and model sequential data makes them more robust and effective at handling complex visual tasks. The transformer is a foundational model that has been driving a paradigm shift in artificial intelligence. It has attracted increasing attention due to its remarkable ability to capture long-range dependencies and model sequential data, and it learns context, and thus, meaning by tracking relationships in sequential data. In transformer models, the self-attention technique is applied to detect the subtle ways even distant data elements in a series influence and depend on each other. Based on these capabilities, the transformer model has been driving a wave of advances in machine learning, and it greatly improves the performance in visual tracking. However, the security of the transformer model in visual tracking has not been thoroughly investigated yet. Although the transformer model has shown impressive performance during many tasks, it is vulnerable to adversarial attacks, which can cause the model to produce incorrect outputs or even fail completely.

In visual tracking, adversarial examples successfully disrupted the performance of tracking algorithms, and they also showed similar attack effects in other deep learning tasks, such as image classification, object detection, and semantic segmentation. Adversarial attacks are particularly concerning in the context of transformer-model-based visual tracking, where the consequences of tracking drift or target loss can be severe. For example, physical attacks [7] can mislead tracking systems by altering traffic signs or the appearance of vehicles, causing autonomous vehicles to make incorrect decisions. Digital attacks [8,9,10,11,12,13,14,15], such as adversarial examples, inject small perturbations into images, preventing trackers from correctly identifying or locating objects. Additionally, environmental manipulations, such as interfering with the image capture capabilities of cameras through lighting and reflection, can effectively disrupt autonomous driving systems. Due to their heavy reliance on correlations and patterns in data, transformers are especially sensitive to fine perturbations. The self-attention mechanism may amplify the effects of these small changes, leading to significant deviations in model output. Studying adversarial examples in deep visual tracking not only reveals its weaknesses but also helps to improve the robustness of algorithms in visual tasks. Our research highlights the security threats faced by transformer models in visual tracking, providing crucial insights for improving existing algorithms and developing more secure deep learning models, thereby enhancing the overall safety and reliability of autonomous driving tracking systems.

In this study, our work aimed to investigate the security of transformer models in visual tracking and evaluate their robustness against different types of adversarial attacks. Specifically, the vulnerability of the transformer models in visual tracking was explored in terms of white-box [13,14,16], gray-box [8,10,11,12], and black-box attacks [9,15]. Moreover, this study analyzed the impact of different attack methods on the tracking performance. The goal of our work was to provide insights into the security of the transformer models in visual tracking and identify potential vulnerabilities that need to be addressed in future research.

Three attacks were deployed in the investigation experiments: a cooling–shrinking attack [8], IoU attack [9], and RTAA attack [13], and the experiments were carried on three data sets: OTB100 [17], VOT2018 [18], and GOT-10k [19]. It is important to note that these three attack methods also have effective impacts on models with CNN, RNN, and LSTM architectures. However, the focus of this study was on the attack effects under the transformer architecture, and therefore, additional descriptions of models with other architectures are not provided.

Figure 1 gives an example: the RTAA attack causes two transformer-model-based trackers to track the wrong targets.

The contributions of this study are summarized as follows:Investigation and analysis: Adversarial attacks against visual tracking tasks were investigated to analyze the tracking principle and the advantages and weaknesses of the transformer-model-based trackers. Moreover, the influences of the adversarial attacks were studied. It is important to direct the design of robust and secure deep-learning-based trackers for visual tracking.Implementation and verification: three adversarial attacks were implemented to perform the attacks on the transformer-model-based visual tracking, and the effectiveness of these attacks was verified on three data sets.

The remaining sections of this paper are as follows: The second part introduces the basic principles of the Transformer model and its application in visual tracking, discussing its sensitivity to adversarial attacks and defense methods to improve robustness. The third part describes in detail the principles of cooling-shrinking attacks, IoU attacks, and RTAA attacks, their impact on the Transformer model, and evaluates these attack methods’ experimental design and results on datasets, analyzing the impact of adversarial attacks on the performance of Transformer models. Finally, the fourth part summarizes the research findings, discusses methods to enhance the security and robustness of Transformer models in visual tracking tasks, and points out future research directions.

## 2. Adversarial Attacks on Transformer-Based Visual Tracking

### 2.1. Transformer Architecture

The transformer model was introduced by Vaswani et al. and applied in machine translation. It is an architecture for transforming one sequence into another with the help of attention-based encoders and decoders. The attention mechanism takes an input sequence into each step and decides at each step whether to facilitate capturing the global information from the input sequence.

The transformer architecture has been used to replace recurrent neural networks in these sequential tasks: natural language processing, speech processing, and computer vision, and has been gradually extended to handle non-sequential problems.

As the important mechanism of transformer architecture, the attention mechanism has been introduced into the tracking field. In [22], Choi et al. adopted channel-wise attention to provide the matching network with target-specific information. It merely borrows the concept of attention to conduct model or feature selection. In [23], Yu et al. explored both self-attention and cross-branch attention to improve the discriminative ability of target features before applying depth-wise cross-correlation. In [24], Du et al. proposed CGACD to learn attention from the correlation result of the template and search region, and then adopted the learned attention to enhance the search region features for further classification and regression. These works improved the tracking accuracy with the attention mechanism, but they still highly rely on the correlation operation in fusing the template and search region features. In [20], Chen et al. designed an attention-based network to directly fuse template and search region features without using any correlation operation.

### 2.2. Transformer Tracking

Transformer tracking is a state-of-the-art object-tracking method that uses the transformer model to achieve accurate and robust object tracking. Compared with traditional object tracking methods, transformer tracking has shown superior performance in handling object deformation and occlusion. The key idea of transformer tracking is to represent each object as a vector learned by the transformer model. During tracking, the feature representation of the object is first converted into a vector and fed into the transformer model for processing, which generates a new representation of the object. The location and state of the object in the next frame are depicted based on the similarity between the old and new representations. Transformer tracking has several advantages over traditional tracking methods. First, the transformer model is capable of capturing the context information of the object, making the tracker more robust to object deformation and occlusion. Second, the representation vector of the object can be dynamically adapted during tracking, which allows the tracker to better adapt to the object’s motion and deformation. Finally, pre-training can be applied to the transformer model to accelerate training and improve the tracking performance.

There have been several recent studies on transformer tracking, such as TransT [20], TMT [25], STARK [26], AiATrack [27], OSTrack [28], SwinTrack [29], TFITrack [30], and TrTr [31]. They utilized the encoder–decoder network to extract the global and rich contextual inter-dependencies. In addition, MixFormer is presented in [21] as a compact tracking framework, and it is built upon transformers. It was proposed to simplify the multi-stage pipeline of feature extraction, target information integration, and bounding box estimation. Moreover, it unifies the process of feature extraction and target information integration.

### 2.3. Adversarial Attacks on Transformer Tracking

Vision is the core and foundation of tracking, and the adversarial robustness of the vision transformer decides the robustness of the tracking with the transformer framework. In recent years, the vision transformer has achieved attention. In [32], the authors showed that standard vision transformer models perform more robustly than standard CNNs under adversarial attacks. In [33], the authors revealed that vision transformer models are not more robust than CNNs if both are trained in the same training framework. It was observed that the accuracy of standard models can be easily reduced to near zero under standard attacks. In addition, Fu et al. [34] studied attacking vision transformer models in a patch-wise approach and revealed the unique vulnerability of vision transformer models. To boost the adversarial robustness of vision transformer models, in [35], the authors explored multiple-step adversarial training of vision transformer models. However, multi-step adversarial training is computationally expensive. To reduce the computational cost, in [36], Wu et al. took the step of exploring fast single-step adversarial training on vision transformer models.

### 2.4. Defense Methods

To enhance the robustness of visual transformer models against adversarial attacks, various defense strategies were proposed. In 2024, Suttapak et al. significantly improved model robustness through multi-step adversarial training by iteratively incorporating adversarial examples during the training process [37]. Input preprocessing techniques, such as image denoising and adaptive contrast enhancement, can effectively reduce the impact of adversarial perturbations. Regularization methods, such as weight decay and dropout, enhance the model stability by preventing overfitting. Frequency-driven defense methods analyze the frequency components of images to identify and filter adversarial perturbations [38]. These methods have demonstrated significant defensive effectiveness across different scenarios and datasets, providing important references for developing more robust visual tracking systems.

## 3. Generating Adversarial Examples

The entire process flow is illustrated in Figure 2. Three attack methods were implemented: cooling–shrinking attack [8], IoU attack [9], and RTAA attack [13]. The cooling-shrinking attack is a generator-based adversarial attack, while the IoU attack and RTAA attack are gradient-based adversarial attacks. These attack methods are applied to the TransT [20] and MixFormer [21] models, respectively, to evaluate the adversarial robustness of their transformer trackers.

### 3.1. Attack Principles

The attack principles of three attack methods are analyzed in detail as follows.

**Cooling–shrinking attack’s principle**: In the cooling–shrinking attack, the proposed adversarial perturbation generator aims to deceive the SiamRPN++ tracker by making the target invisible and leading to tracking drift. This is achieved by training the generator with a cooling–shrinking loss. The generator is designed to attack either the search regions or the template, where the search region is the located target, and the template is given in the initial frame.

The designed cooling–shrinking loss is composed of the cooling loss $L_{C}$ to interfere with the heat maps $M_{H}$, and the shrinking loss $L_{S}$ interferes with the regression maps $M_{R}$, where the heat maps $M_{H}$ and the regression maps $M_{R}$ are important components of the SiamRPN++ tracker.

In the generator, the cooling loss $L_{C}$ is designed to cool down the hot regions that the target may focus on, causing the tracker to lose the target, and the shrinking loss $L_{S}$ is designed to force the depicted bounding box to shrink, leading to error accumulation and tracking failure.

**IoU attack’s principle**: The IoU attack method aims to decrease the IoU scores between the depicted bounding boxes and ground truth bounding boxes in a video sequence, indicating the degradation of the tracking performance. It is designed to counter existing black-box adversarial attacks that target static images for image classification. Unlike the existing black-box adversarial attacks, the IoU attack generates perturbations by considering the depicted IoU scores from both the current and previous frames. By decreasing the IoU scores, the IoU attack reduces the frame-by-frame accuracy of coherent bounding boxes in video streams. During the IoU attack, learned perturbations are utilized and transferred to subsequent frames to initiate a temporal motion attack. In the IoU attack, there is an increase in the noise level as the IoU scores decreases, but this relationship is not linear: in an IoU attack, a clean input frame is subjected to the addition of heavy uniform noise, resulting in a heavily noised image with a low IoU score. During the addition process, the IoU scores gradually decline as the noise level increases.

The following employed strategy achieves the effectiveness and imperceptibility of the IoU attack in video streams: there exists a positive correlation between the direction of decrease in IoU and the direction of increase in noise. However, this relationship is not linear. The IoU attack gradually reduces the IoU score for each frame in a video stream by adding the minimum amount of noise. It identifies the specific noise perturbation that results in the lowest IoU score among an equal amount of noise levels through orthogonal composition.

**RTAA attack’s principle**: The RTAA attack takes temporal motion into consideration over the estimated tracking results frame-by-frame.

The RTAA attack creates a pseudo-classification label and a pseudo-regression label, and both labels are used to design the adversarial loss. The adversarial loss is set to make $L_{c}$ and $L_{r}$ be the same when correct and pseudo-labels are used separately, where $L_{c}$ denotes the binary classification loss and $L_{r}$ is the bounding box regression loss; these are two important parameters in deep visual tracking algorithms.

In deep visual tracking, the binary classification loss is a measure used to evaluate the performance of a visual tracking algorithm. Visual tracking is often framed as a binary classification problem, where the goal is to distinguish between the target and the background. The binary classification loss function in visual tracking measures the difference between the depicted class probabilities and the true class labels. In this case, the two classes are the target and the background. The loss function is used to train the visual tracking algorithm and adjust its parameters so that it improves its ability to accurately track the target over time. Moreover, the bounding box regression loss in visual tracking is a measure used to evaluate the performance of a visual tracking algorithm in depicting the location and the size of the bounding box that encloses the target. In visual tracking, the goal is to track the target of interest over time, and the bounding box regression loss function is used to adjust the parameters of the tracking algorithm so that it can accurately depict the location and size of the bounding box that encloses the target in each frame of the video sequence.

### 3.2. Advantages and Weaknesses of Attacks

**Cooling–shrinking attack’s advantages**: There are two advantages: (i) the use of a cooling–shrinking loss allows for fine-tuning of the generator to generate imperceptible perturbations while still effectively deceiving the tracker, and (ii) the method is able to attack the SiamRPN++ tracker, which is currently one of the most powerful trackers, as seen by achieving the state-of-the-art performance on almost all tracking data sets.

**Cooling–shrinking attack’s weaknesses**: There are three weaknesses: (i) the method is specifically designed to attack the SiamRPN++ tracker and may not be effective against other types of trackers; (ii) the generator is trained with a fixed threshold, and thus, it may not be effective against different scenarios or environments; and (iii) the attack method may have limited use in real-world applications, such as adding adversarial perturbations to targets being tracked.

**IoU attack’s advantages**: There are three advantages: (i) the IoU attack involves both spatial and temporal aspects of target motion, making it more comprehensive and challenging for visual tracking; (ii) the method uses a minimal amount of noise to gradually decrease the IoU scores, making it more effective in terms of computational costs; and (iii) the IoU attack can be applied to different trackers as long as they depict one bounding box for each frame, which makes it more versatile.

**IoU attack’s weaknesses**: There are three weaknesses: (i) the exact relationship between the noise level and the decrease of IoU scores is not explicitly modeled, making it difficult to optimize the noise perturbations; (ii) the method involves a significant amount of computation during each iteration, which might affect its efficiency in real-world applications; and (iii) the method relies on the assumption that the trackers use a single bounding box prediction for each frame, which might not always be the case in some complex scenarios.

**RTAA attack’s advantages**: There are three advantages: (i) the RTAA attack generates adversarial perturbations based on the input frame and the output response of deep trackers, which makes the adversarial examples more effective and realistic; (ii) the attack uses the tracking-by-detection framework, which is widely used in computer vision tasks and helps to increase the robustness of the attack; and (iii) the method can effectively confuse the classification and regression branches of the deep tracker, which results in rapid degradation in performance.

**RTAA attack’s weaknesses**: There are four weaknesses: (i) the method relies on a fixed weight parameter λ, which may not be optimal for different types of deep trackers and attack scenarios; (ii) the method uses a random offset and scale variation for the pseudo-regression label, which may not be effective for all tracking scenarios; (iii) the method requires multiple iterations to produce the final adversarial perturbations, which increases the computational complexity of the attack; and (iv) the method considers the adversarial attacks in the spatiotemporal domain, which may limit its applicability to other computer vision tasks that do not have a temporal aspect.

### 3.3. Transformer Tracking Principles

**TransT’s principle**: Correlation plays an important role in tracking. However, the correlation operation is a local linear matching process, which easily leads to losing semantic information and falls into a local optimum. To address this issue, inspired by the transformer architecture, TransT [20] was proposed with the attention-based feature fusion network, and it combines the template and search region features solely using an attention-based fusion mechanism.

TransT consists of three components: a backbone network, a feature fusion network, and a prediction head. The backbone network extracts the features of the template and the search region, separately. With the extracted features, then, the features are enhanced and fused by the proposed feature fusion network. Finally, the prediction head performs the binary classification and bounding box regression on the enhanced features to generate the tracking results.

**MixFormer’s principle**: To simplify the multi-stage pipeline of tracking and unify the process of feature extraction and target information integration, a compact tracking framework is proposed in [21], termed as MixFormer, which was built upon transformers.

MixFormer utilizes the flexibility of attention operations and uses a mixed attention module for simultaneous feature extraction and target information integration. This synchronous modeling scheme allows for extracting target-specific discriminative features and performs the extensive communication between the target and search areas. MixFormer simplifies the tracking framework by stacking multiple mixed attention modules with embedding progressive patches and placing a localization head on top. In addition, to handle multiple target templates during online tracking, an asymmetric attention scheme was designed in the mixed attention module to reduce the computational cost, and an effective score prediction module was proposed to select high-quality templates.

### 3.4. Investigation Experiments and Analyses

Investigation experiments evaluated the robustness of tracker models based on the transformer framework, namely, TransT and MixFormer, against three distinct adversarial attack methods, and the evaluation was performed on three foundational benchmark datasets: OTB2015 [17], VOT2018 [18], and GOT-10k [19]. The investigated attack methods encompass a white-box attack (RTAA attack), semi-white-box attack (CSA attack), and black-box attack (IoU attack). The objective was to comprehensively assess the vulnerability of these trackers under varying degrees of adversarial perturbations, shedding light on their limitations and potential defense strategies. The findings from this study contributed to enhancing the overall reliability and security of transformer-based trackers in real-world scenarios.

Standard evaluation methodologies were adopted for the benchmark datasets. For the OTB2015 [17] dataset, the one-pass evaluation (OPE) was utilized, which employs two key metrics: a precision curve and a success curve. The precision curve quantifies the center location error between the tracked results and the ground truth annotations, and is computed using a threshold distance, such as 20 pixels. The success curve measures the overlap ratio between the detected bounding boxes and the ground truth annotations, reflecting the accuracy of the tracker at different scales.

This study evaluated object tracking algorithms on the VOT2018 [18] dataset using accuracy, robustness, failures, and expected average overlap (EAO) as the evaluation metrics. Accuracy measures the precision of tracking algorithms in predicting the target’s position, while robustness assesses the algorithm’s resistance to external disturbances. The failures count the number of times the tracking process fails, and the expected average overlap provides a comprehensive metric considering both accuracy and robustness, which is calculated by integrating the success rate curve to evaluate the overall performance of the object tracking algorithms.

The average overlap (AO) and success rate (SR) were adopted as evaluation metrics for the GOT-10k [19] dataset. The average overlap measures the average degree of the overlap between the tracking results and the ground truth annotations and reflects the accuracy of the tracker’s predictions regarding the target’s locations. The success rate assesses the success detection rate of the tracker at specified thresholds, where the thresholds were set at 0.5 and 0.75. ${\mathrm{SR}}_{0.5}$ and ${\mathrm{SR}}_{0.75}$ represent the success rates with overlaps greater than 0.5 and 0.75, respectively. A higher SR value indicates that the tracker successfully detected the target within a larger overlapping range.

In Table 1, the precision is a measure of accuracy, and it is calculated as shown in Equation (Equation 1).
(1)$$Precision=\frac{1}{f}\sum_{i=1}^{f}p\left(i\right).$$

The precision is calculated by taking the reciprocal (1 divided by) of the average center location error across all frames. Each frame’s center location error represents how far off the predicted bounding box’s center is from the ground truth bounding box’s center. This error is found by calculating the Euclidean distance between these two centers for each frame. The precision is obtained by adding up these errors for all frames and then dividing by the total number of frames (denoted as “*f*”).

The success measures how well the predicted bounding box overlaps with the ground truth bounding box. To calculate the success, the reciprocal (1 divided by) of the average overlap degree is taken across all frames. The overlap degree for each frame is determined by dividing the area of intersection between the predicted bounding box and the ground truth bounding box by the area of their union. The success metric is calculated by adding up these overlap degrees for all frames and then dividing by the total number of frames (“*f*”).

In the dataset VOT2018 [18], visual attributes (e.g., partial occlusion, illumination changes) were annotated for each sequence to evaluate the performance of trackers under different conditions. An evaluation system should detect errors (failures) when a tracker loses the track and re-initialize the tracker after five frames following the failure for effectively utilizing the dataset. Five frames for the re-initialization were chosen because immediate initialization after failure leads to subsequent tracking failures. Additionally, since occlusions in videos typically did not exceed five frames, this setting was established. It is a distinctive mechanism to enable “reset” or “re-initialize”, where a portion of frames after the reset cannot be used for evaluation.

In Table 2, the accuracy metric evaluates how well the predicted bounding box (refer to as $A_{t}^{T}$) aligns with the ground truth bounding box (refer to as $A_{t}^{G}$) for a given frame in a tracking sequence, denoted as the tth frame. This accuracy metric is symbolically represented as $\phi_{t}$. Furthermore, $\phi_{t}\left(i,k\right)$ represents the accuracy of the tth frame within the kth repetition of a particular tracking method, where the total number of repetitions is indicated as $N_{rep}$. To calculate the average accuracy for this specific tracking method (ith tracker), the mean accuracy over all valid frames ($N_{valid}$), $\rho_{A}\left(i\right)$, needs to be determined: $\rho_{A}\left(i\right)$ is computed as the sum of all $\phi_{t}\left(i\right)$ values divided by the total number of valid frames, $N_{valid}$, where *t* ranges from 1 to $N_{valid}$. The robustness, conversely, gauges how stable a tracking method is when following a target, and a higher robustness value indicates a lower level of stability. The robustness is quantified by using the following mathematical expression: $\rho_{R}\left(i\right)$ is calculated as the sum of tracking failures F(i,k) in the kth repetition of the ith tracking method, divided by the total number of repetitions, $N_{rep}$. In Table 3, the “Failures” index counts the instances of tracking failures that occur during the tracking process of a tracking algorithm. These failures are typically related to tracking errors and do not include specific restarts or skipped frame numbers.

In Table 3, the expected average overlap (EAO) is denoted as $\phi_{N_{s}}$. This metric is designed to quantify the expected average coverage rate specifically for tracking sequences up to an intended maximum length ($N_{s}$). To compute the EAO, the average intersection over union (IoU) value is considered, denoted as $\phi_{i}$, for frames ranging from the first frame to the $N_{s}$th frame in the sequence, even including the frames where tracking may have failed, and $N_{s}$ represents the total sequence length. In the context of the VOT2018 [18] dataset, the calculation of the expected average overlap involves taking the average of the EAO values within an interval $\left[N_{low},N_{high}\right]$, which corresponds to typical short-term sequence lengths, and the expected average overlap is denoted as $\hat{\phi }$ and is calculated by Equation (Equation 2).
(2)$$\hat{\phi }=\frac{1}{N_{high}-N_{low}}\sum_{N_{s}=N_{low}:N_{high}}{\hat{\phi }}_{N_{s}},$$
where the $N_{s}$ ranges from $N_{low}$ to $N_{high}$, and the $\hat{\phi }$ captures the expected average overlap across a range of sequence lengths, providing valuable insights into the tracking performance.

In Table 4, a metric called the average overlap (AO) is utilized to gauge the extent of the overlap that occurs during the tracking process. The AO is determined by assessing the degree of overlap for each individual frame and subsequently computing the average of these individual overlaps. The AO is the average level of the overlap, and it takes the sum of the overlap values for each frame, and then it is divided by the total number of frames (*N*) in the sequence. Each “$Overlap_{i}$” represents the extent of overlap for the ith frame. Additionally, Table 4 and Table 5 employ a metric known as the success rate (SR) to assess how well the tracker performs under various overlap threshold conditions, and the SR quantifies the ratio of frames in which the tracker successfully keeps track of the target while considering a specific overlap threshold. The SR is a measure of how effectively the tracker follows the target. To compute it, an indicator function (*I*) applied to each frame’s overlap value is summed up. If the overlap ($Overlap_{i}$) is greater than or equal to the specified threshold (Threshold), *I* equals 1; otherwise, it equals 0. The resulting sum is then divided by the total number of frames (*N*) in the sequence. For example, $SR_{0.5}$ refers to the scenario where the overlap threshold is set to 0.5, and $SR_{0.75}$ corresponds to a threshold of 0.75. These metrics offer valuable insights into how well the tracking system performed at different levels of overlap.

The experimental results are shown as follows. **Results on the dataset OTB2015** (shown in Table 1 and Figure 3):

The original results shown in Table 1 and Figure 3, along with the results under three types of adversarial attacks, were compared. It was observed that all three attacks had certain impacts. In terms of the success rate and precision, the white-box attack RTAA performed the best, causing decreases of 93.2% and 97.4% in the success rate and drops of 94.5% and 95.7% in the precision for MixFormer and TransT, respectively. The next was the black-box attack IoU, which resulted in success rate decreases of 20.3% and 9.4% and precision decreases of 18.4% and 4.6% for MixFormer and TransT, respectively. Finally, the impact of the semi-black-box attack CSA, trained by SiamRPN++, was the least pronounced, with a minimal influence on the tracking results. When attacking the MixFormer and TransT models, they were based on the transformer framework, and their success rates dropped by 8.0% and 4.2% and their precision values decreased by 7.6% and 3.2%, respectively.

**Results on the dataset VOT2018** (shown in Table 2 and Table 3 and Figure 4):

As shown in Table 2, the RTAA attack achieved the best performance, followed by the IoU attack, and the CSA attack had the lowest effectiveness. Specifically, both trackers’ accuracies were significantly reduced after being subjected to adversarial attacks, indicating a noticeable deviation between the tracking results after adversarial attacks and the original results. In Table 3, ranked in the order of the RTAA, IoU, and CSA adversarial attacks, the main metric EAO scores for MixFormer decreased by 96.1%, 38.9%, and 10%, respectively, while for TransT, they decreased by 95.4%, 47%, and 0%.

Figure 4 presents the performances of different attributes on the VOT2018 [18] dataset, comparing the tracking results under three types of adversarial attacks with the original results in various specific scenarios. In the radar chart, the closer a point is to the center, the worse the algorithm performed on the attribute, while points farther from the center indicate better performance.

Upon observing the target radar chart on the VOT2018 [18] dataset, a decline in tracking performance was clear when facing the three types of adversarial attacks, including scenarios involving occlusion, unassigned, and overall. Among them, the RTAA attack had the strongest effect, as it exhibited nearly the worst performance in all scenarios, where the preselected box did not cover the tracking target. The IoU attack came next, showing a comprehensive performance decrease across all scenarios. As for the CSA attack, it exhibited enhancement in certain scenarios because the CSA attack mainly targeted the SiamRPN++ model and exhibited significant attack effectiveness on this model. This means that the transferability of the CSA attack was not good for the TransT and MixFormer models.

**Results on the dataset GOT-10k** (shown in Table 4 and Table 5, and Figure 5):

As shown in Table 4 and Table 5, and Figure 5, three types of adversarial attacks on both trackers were conducted on the GOT-10k [19] dataset. By observing the metrics of average overlap (AO), success rate at 0.5 overlap ($SR_{0.5}$), and success rate at 0.75 overlap ($SR_{0.75}$), it was evident that the overall performance of these trackers decreased. Specifically, the MixFormer and TransT trackers experienced declines in the average overlap (AO) of 93.3% and 22.6%, 5.0% and 93.6%, and 26.5% and 2.5% under the RTAA attack, the IoU attack, and the CSA attack, respectively.

## 4. Conclusions

This study conducted an in-depth investigation on the performance and vulnerabilities of transformer-based visual trackers when facing adversarial attacks. Through extensive experimental evaluations, we revealed the significant impact of various types of adversarial attacks—including white-box, black-box, and semi-white-box attacks, especially white-box RTAA attacks—on reducing the tracking performance of transformer models. The experimental results show that even state-of-the-art transformer trackers experienced significant performance degradation when encountering carefully crafted adversarial examples. Moreover, our research emphasized the importance of ensuring the security and robustness of transformer-based visual tracking systems during development and deployment. Future work should focus on testing these improved methods and mechanisms in real-world applications to assess their effectiveness and feasibility in actual environments. We will validate the effectiveness of attacks in diverse scenarios, analyze the real-world impact of adversarial attacks, and simultaneously design more effective defense strategies for specific attack methods. Additionally, we will optimize existing defense mechanisms, such as adversarial training and input preprocessing, to enhance their effectiveness in resisting adversarial attacks.

## Figures and Tables

**Figure 1 sensors-24-04761-f001:**
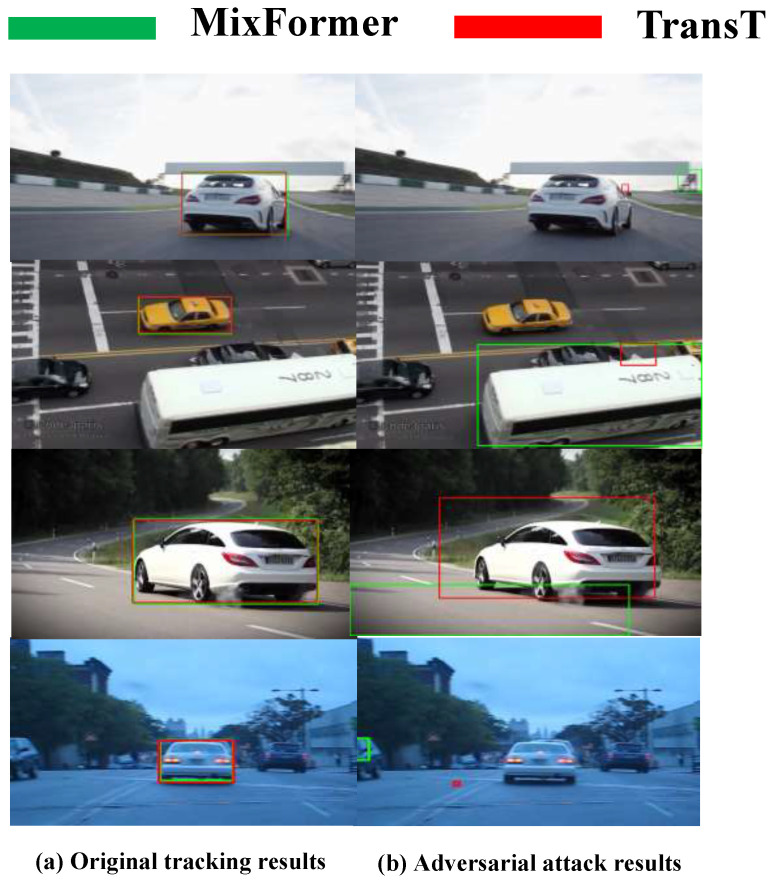
The adversarial attack, RTAA, in two transformer-model-based trackers (TransT [20] and MixFormer [21]). The TransT tracker effectively located targets in the original video sequences. The MixFormer utilized the flexibility of attention operations, and there was a mixed attention module for simultaneous feature extraction and target information integration. The original result of the tracker as shown in (**a**), The adversarial attack strategy decreased the tracking accuracy, as shown in (**b**), with the RTAA attack, i.e., the TransT and MixFormer trackers output incorrect bounding boxes to track the wrong targets.

**Figure 2 sensors-24-04761-f002:**
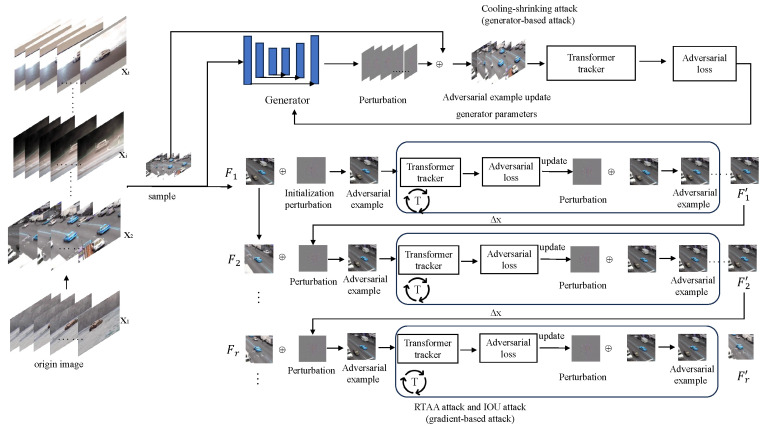
The adversarial attack flowchart for Transformer trackers can be divided into two categories: gradient descent based attacks and generator based attacks, which include three types of attacks: cooling-shrinking attacks, IOU attacks, and RTAA attacks. In the attack section based on gradient descent in the figure, Deltax represents perturbation interpolation between frames, and *T* represents the number of iterations.

**Figure 3 sensors-24-04761-f003:**
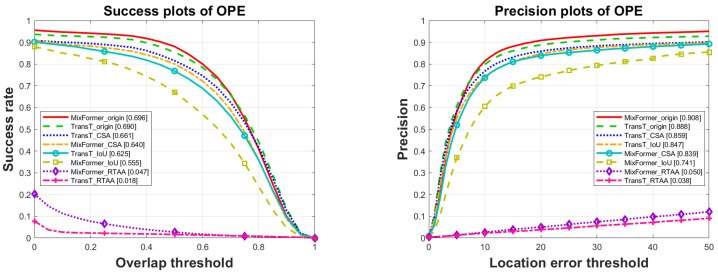
Evaluation results of trackers with and without adversarial attacks on the dataset OTB2015.

**Figure 4 sensors-24-04761-f004:**
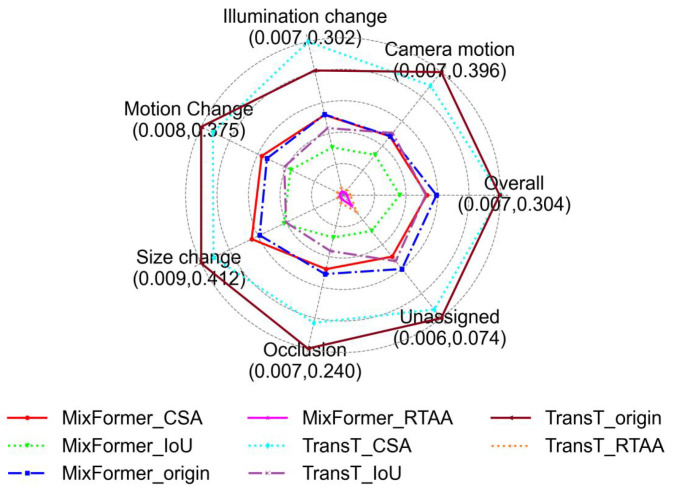
Quantitative analysis of different attributes on the dataset VOT2018.

**Figure 5 sensors-24-04761-f005:**
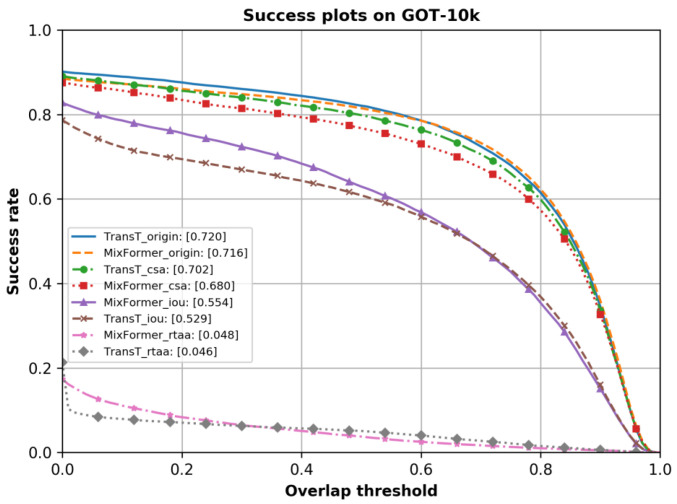
Evaluation results of trackers with or without adversarial attacks on the dataset GOT-10k.

**Table 1 sensors-24-04761-t001:** Attack performance on the dataset OTB2015.

Tracker	Success				Precision			
	**Original**	**Attack_CSA**	**Attack_IoU**	**Attack_RTAA**	**Original**	**Attack_CSA**	**Attack_IoU**	**Attack_RTAA**
MixFormer	0.696	0.640	0.555	0.047	0.908	0.839	0.741	0.050
TransT	0.690	0.661	0.625	0.018	0.888	0.859	0.847	0.038

**Table 2 sensors-24-04761-t002:** Attack performance on the dataset VOT2018 (accuracy and robustness).

Tracker	Accuracy				Robustness			
	**Original**	**Attack_CSA**	**Attack_IoU**	**Attack_RTAA**	**Original**	**Attack_CSA**	**Attack_IoU**	**Attack_RTAA**
MixFormer	0.614	0.625	0.599	0.198	0.698	0.819	1.288	10.339
TransT	0.595	0.592	0.578	0.111	0.337	0.323	0.899	5.984

**Table 3 sensors-24-04761-t003:** Attack performance on the dataset VOT2018 (failures and EAO).

Tracker	Failures				EAO			
	**Original**	**Attack_CSA**	**Attack_IoU**	**Attack_RTAA**	**Original**	**Attack_CSA**	**Attack_IoU**	**Attack_RTAA**
MixFormer	149	175	275	2208	0.180	0.162	0.110	0.007
TransT	72	69	192	1278	0.302	0.304	0.160	0.014

**Table 4 sensors-24-04761-t004:** Attack performance on the dataset GOT-10k (AO (%) and ${\mathrm{SR}}_{0.5}$ (%)).

Tracker	AO (%)				${\mathrm{SR}}_{0.5}$ (%)			
	**Original**	**Attack_CSA**	**Attack_IoU**	**Attack_RTAA**	**Original**	**Attack_CSA**	**Attack_IoU**	**Attack_RTAA**
MixFormer	0.716	0.680	0.554	0.048	0.815	0.768	0.629	0.037
TransT	0.720	0.702	0.529	0.046	0.821	0.798	0.609	0.051

**Table 5 sensors-24-04761-t005:** Attack performance on the dataset GOT-10k (${\mathrm{SR}}_{0.75}$ (%)).

Tracker	${\mathrm{SR}}_{0.75}$ (%)			
	**Original**	**Attack_CSA**	**Attack_IoU**	**Attack_RTAA**
MixFormer	0.687	0.633	0.428	0.013
TransT	0.680	0.661	0.433	0.021

## Data Availability

Data are contained within the article.

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
