# Peer review of "Security in Transformer Visual Trackers: A Case Study on the Adversarial Robustness of Two Models"

_sensors, 2024, doi:10.3390/s24144761_

Round 1

Reviewer 1 Report

Comments and Suggestions for Authors

The paper makes significant contributions to the field of AI security, particularly in the context of visual tracking with transformers. It highlights critical vulnerabilities and sets the stage for further research into robust machine learning models. However, the focus is predominantly diagnostic, and future work should aim at developing comprehensive solutions to these identified vulnerabilities. The findings are robust and scientifically sound, but additional work on generalizability and defensive strategies would enhance the paper's impact and practical relevance.

While the introduction provides a general background on the importance of security in visual trackers for autonomous driving, it could benefit from a broader literature review on the specific vulnerabilities of transformer models compared to other architectures. Including a few more recent and relevant references would enrich the context and show a clearer gap that this research aims to fill.

The choice of focusing on transformer models is justified given their relevance and recent adoption in visual tracking tasks. However, the design could be improved by including comparisons to non-transformer models to benchmark the adversarial robustness across different architectures. This would provide a more comprehensive understanding of whether the vulnerabilities identified are unique to transformers.

The methods section details the adversarial attacks implemented, but it lacks comprehensive coverage of the defensive mechanisms tested or any mitigation strategies proposed. Detailing these aspects would strengthen the methodological approach and provide a more balanced view of attack vs. defense dynamics in transformer visual trackers.

The results are robust and well-presented with clear visualizations and statistical evidence. However, the results section could be improved by including more detailed discussions on the implications of these results in practical scenarios, such as how these vulnerabilities might affect real-world deployments of autonomous driving systems. 

Figures are very low in quality and give out a weird sense. 

All Figures must be generated again with higher quality and consistency in their creation. 

Comments on the Quality of English Language

English is fine. 

Reviewer 2 Report

Comments and Suggestions for Authors

Realtime object detection and recognition is still an evolving area of research.

1.      In abstract, state the specific techniques used and the significance of the obtained results (eg. Highest performance value)

2.      Discuss the impact of adversarial attacks, on auto drive systems.

3.      References should be cited appropriately. For example, in 1st paragraph there are no any references to support the facts. However, in 2nd para some of the references are not directly applicable to the said study domain.  Also, in line 66, the reference is not included. Likewise, refine the paper with proofread.  Most of the facts in Section 2 should be supported by references.

4.      Structure should be named from 1. Introduction. …  Refine the section numbering.

5.      The information flow of the entire paper is not clear. It would be better to include a diagram showing the organization of the paper.

6.      What is the novel scientific contribution of the paper.

7.      A discussion section should be included, that compares the results with the existing studies. By that, you can justify the novel contribution of your work.

8.      Any practical applications, that you can deploy your proposed model?  Any case studies to try out in the real-time?

Comments on the Quality of English Language

Paper should be proofread

Round 2

Reviewer 1 Report

Comments and Suggestions for Authors

Thank you for taking into account your comments.

The manuscript now has higher quality. 

Reviewer 2 Report

Comments and Suggestions for Authors

Realtime object detection and recognition is still an evolving area of research.

1.      Still the facts mentioned in the 1st paragraph need references.

2.      Include a complete process flow diagram and explain in the methodology.

3.      What is the novel scientific contribution of the paper.

4.      A discussion section should be included, that compares the results with the existing studies. By that, you can justify the novel contribution of your work.

5.      Discuss how you achieved the defined research questions of this study, by referring to the methodology you followed and the results obtained.

6.      Discuss the usefulness of the proposed approach.

7.      Discuss the limitations of your study and future possible extensions.

8.      Any practical applications, that you can deploy your proposed model?  Any case studies to try out in the real-time?

Comments on the Quality of English Language

Proofread the paper
